# Strain-Specific Biostimulant Effects of *Chlorella* and *Chlamydomonas* Green Microalgae on *Medicago truncatula*

**DOI:** 10.3390/plants10061060

**Published:** 2021-05-25

**Authors:** Margaret Mukami Gitau, Attila Farkas, Benedikta Balla, Vince Ördög, Zoltán Futó, Gergely Maróti

**Affiliations:** 1Institute of Plant Biology, Biological Research Center, 6726 Szeged, Hungary; margaret_mukami@yahoo.com (M.M.G.); farkas.attila@brc.hu (A.F.); balla.benedikta@brc.hu (B.B.); 2Department of Plant Sciences, Faculty of Agricultural and Food Sciences, Széchenyi István University, 9246 Mosonmagyaróvár, Hungary; ordog.vince@sze.hu; 3Research Centre for Plant Growth and Development, School of Life Sciences, Campus Pietermaritzburg, University of KwaZulu-Natal, 3201 Scottsville, Pietermaritzburg, South Africa; 4Department of Irrigation Development and Melioration, Hungarian University of Agriculture and Life Sciences, 5540 Szarvas, Hungary; Futo.Zoltan@uni-mate.hu; 5Department of Water Sciences, University of Public Service, 6500 Baja, Hungary

**Keywords:** microalgae, *Chlorella*, *Chlamydomonas*, *Medicago truncatula*, plant growth, biostimulant

## Abstract

Microalgae have been identified to produce a plethora of bioactive compounds exerting growth stimulating effects on plants. The objective of this study was to investigate the plant-growth-promoting effects of three selected strains of eukaryotic green microalgae. The biostimulatory effects of two *Chlorella* species (MACC-360 and MACC-38) and a *Chlamydomonas reinhardtii* strain (cc124) were investigated in a *Medicago truncatula* model plant grown under controlled greenhouse conditions. The physiological responses of the *M. truncatula* A17 ecotype to algal biomass addition were characterized thoroughly. The plants were cultivated in pots containing a mixture of vermiculite and soil (1:3) layered with clay at the bottom. The application of live algae cells using the soil drench method significantly increased the plants’ shoot length, leaf size, fresh weight, number of flowers and pigment content. For most of the parameters analyzed, the effects of treatment proved to be specific for the applied algae strains. Overall, *Chlorella* application led to more robust plants with increased fresh biomass, bigger leaves and more flowers/pods compared to the control and *Chlamydomonas*-treated samples receiving identical total nutrients.

## 1. Introduction

The human population has significantly increased over the past few decades. This increase has raised the demand for food but reduced land for crop production because of urbanization and clearance for human settlement. To increase food supply, chemical fertilizers have been used for various crops including forage and food crops. Unfortunately, chemical fertilizers have detrimental effects on the environment as they cause accumulation of nitrogen and phosphorous to harmful levels. The accumulated nutrients alter the balance in most ecosystems and hence reduce biodiversity. Additionally, chemical fertilizers no longer have a significant impact on crop yield unless when used at high concentrations, which is not economical for farmers.

Since the global demand for food is projected to double by 2050 [1], it is imperative to find alternatives to increase crop production. There is a need for novel agricultural methods that promote sustainable use of natural resources, such as salty water, and allow reclamation of polluted lands. Biostimulants from seaweeds [2,3,4,5,6,7,8,9], plants [10,11,12] and microorganisms (monocultures and co-cultures) [13,14,15,16,17,18] have been found to improve plant growth and yield in several plant species under normal as well as stressful conditions. However, preparation of these biostimulants requires high energy and is labor intensive in most cases. While seaweeds are abundant in the seas, continuous harvesting alters the water ecosystems, and at some point the seaweed population might be depleted. Plants, on the other hand, require fertile land for cultivation and may take a long time to grow sufficient biomass for biostimulant processing. In contrast, microorganisms multiply rapidly but may require special media and facilities for mass cultivation and biomass processing.

Microalgae (MA) represent a group of photosynthetic microorganisms that grow rapidly not only in clean but also in wastewater in natural association with bacteria and fungi. Studies have shown that most eukaryotic green microalgae produce a plethora of bioactive compounds with a wide range of applications in several sectors such as animal feed, human food, pharmaceuticals, aquaculture and hydroponic crop production [19,20,21,22,23]. Thus, MA represent a viable alternative for biostimulant/biofertilizer production since they can be incorporated into various systems such as wastewater management or aquaculture and hydroponic crop production. To increase the environmental impact of MA use in agriculture, live cell suspension or whole algae cultures cultivated in wastewater can be used for application to soil (or plants) [10,24,25], eliminating the need for expensive clean water and mineral addition for farming. This process also eradicates energy expenditure in extraction procedures or wastewater management. Algae biomass can be concentrated by flocculation (using bacterial or fungal co-culture, which also promotes biomass accumulation) [26,27,28]. Overall, using live MA cells in agriculture saves water, chemicals/fertilizers, energy, time and space.

*Medicago truncatula* is a model plant of the Fabaceae family, the third largest angiosperm family, which, after the Gramineae (Poaceae) family, is the most important to humans [29]. Several commercial crops such as soybean, garden pea, peanut and alfalfa, the world’s most cultivated and most valuable forage plant, belong to this family. These plants can fix nitrogen and are, therefore, important sources of oil and protein for animals and humans. In addition, they sequester carbon, which makes them promising candidates for fuelwood. Although most studies on legumes focus on their interaction with plant-growth-promoting rhizobacteria (PGPR), a few studies evaluate the effect of seaweed and MA biostimulants on these plants including *Phaseolus vulgaris* [7,30], *Vigna radiata* [6,31,32], *Glycine max* [33,34] and *Medicago sativa* [35,36,37]. In plant biostimulant studies, MA were administered to plants in the form of extracts [32,38,39,40], dry biomass [41,42,43,44,45], spent medium/supernatant [46,47], whole cultures [46] as well as cell suspensions [47,48], and desirable results were achieved. Application of the same MA strain’s dry biomass, liquid fertilizer and foliar application all led to positive results, although foliar application had greater effects [49]. Overall, the desirable results relative to the controls were observed irrespective of the application method of MA. This implies that MA indeed had positive effects on plants, although the effects may be dependent on species, cultivar, concentration and mode of application [29]. Monocultures [41] as well as co-cultures of eukaryotic green algae with other green algae, bacteria, cyanobacteria, fungi and all together have all shown plant growth stimulation [34,50,51,52,53,54]. However, to the best of our knowledge, no study sought to elucidate the effects of axenic monocultures of *Chlorophyta* microalgae without any other accompanying microbes on *M. truncatula* plants.

In most studies on the effect of MA on plants, *Chlorella* [42,47,48,49,55,56,57,58,59], *Scenedesmus* [32,34,39] and the cyanobacterium *Arthrospira* (*Spirulina*) [60,61,62] species have been investigated for their potential biostimulatory effect on a whole range of plants including corn, spinach, Chinese chives, onions, lettuce and tomatoes. *Chlorella* is the most popular because of its rapid growth and ability to thrive in a wide range of environmental conditions including adverse ones such as drought, saline, cold and hot habitats. In contrast, *Chlamydomonas* species remain unstudied and underutilized in agriculture despite being some of the most abundant microalgae species in natural soil ecosystems. This is despite the species’ rapid biomass accumulation and capacity to produce phytohormones such as auxins, ethylene, brassinosteroids, cytokinin and trehalose [63,64,65,66].

To fill the above identified gaps in research on MA as biostimulants, our study aimed to investigate the specific effects of selected green eukaryotic MA on *M. truncatula* grown under controlled greenhouse conditions. We conducted a comparative study of the growth-promoting effects of two *Chlorella* strains and one *Chlamydomonas* strain on *M. truncatula* when administered as live cells via the soil drench method. Our main focus was on parameters that additively determined yield and quality. These included plant structure/morphology, height, flower number, biomass and pigment content.

## 2. Results

### 2.1. Characterization of the Selected Microalgae Strains

#### 2.1.1. Algae Growth and Morphology

All investigated green algae strains had reached a stationary phase by the 5th day according to optical density measurements (Figure 1a).

However, the cell numbers indicated that *Chlorella* MACC-360 had reached a plateau much earlier than the two other strains. *Chlorella* MACC-360 also had the highest number of cells, approximately 8-fold that of *C. reinhardtii* cc124 and 3-fold that of *Chlorella* MACC-38 (Figure 1b). The differences in cell numbers are attributable to the fact that *Chlorella* MACC-360 had a higher initial cell number value for the same optical density owing to its small cell size. In addition, *Chlorella* MACC-360 has a shorter cell division cycle compared to the other strains.

Scanning electron microscopy was applied to investigate the morphology and size of the applied green microalgae cells. The pictures revealed that the three strains were different with regards to size and surface texture. *Chlorella* MACC-38 (Figure 2a) had a rough surface, while *C. reinhardtii* cc124 and *Chlorella* MACC-360 appeared smooth (Figure 2b–d).

*C. reinhardtii* cc124 had the largest size followed by *Chlorella* MACC-38, and *Chlorella* MACC-360 was the smallest. Neither extracellular material nor cellular aggregations were present in *Chlorella* MACC-38 and *C. reinhardtii* cc124 cultures. In contrast, strong aggregations appeared in *Chlorella* MACC-360; the cells in these aggregations were joined to each other by filament-like extracellular material.

#### 2.1.2. Extracellular Polysaccharide Production of the Selected Microalgae

Confocal laser scanning microscopy (CLSM) was used to investigate the potential production of the extracellular matrix. Seven-day-old MA cells were stained with calcofluor white (CFW) and concanavalin A (Con A) dyes (Figure 3). CFW dye binds to the β-D glucopyranose polysaccharides, while Con A binds to the α-D glucopyranose polysaccharides [67,68].

*C. reinhardtii* cc124 was not only larger in cell size than *Chlorella* species, but it was also different with respect to cell wall composition since CFW dye did not stain the *Chlamydomonas* cells. *Chlorella* species MACC-38 and MACC-360 had similar cell wall composition indicated by the blue fluorescence (Figure 3a,c, respectively). However, *Chlorella* MACC-38 was significantly larger in cell size than *Chlorella* MACC-360. In addition, MACC-38 did not form strong aggregates like *Chlorella* MACC-360 (Figure 3c).

Both *Chlorella* species were stained with dyes specific for polysaccharides. However, *Chlorella* MACC-38 produced polysaccharides that are localized in the cell wall, while *Chlorella* MACC-360 produced polysaccharides, which were secreted out of the cells. The CLSM pictures confirmed that the extracellular material observed under the scanning electron microscope was extracellular polysaccharides.

The most striking difference was the ability of *Chlorella* MACC-360 to produce EPS (green fluorescence from the third day (Appendix A, 360 C) of inoculation with the signal getting stronger with time as shown on day 5 (Figure 3c and Appendix A, 360 E). This implies that *Chlorella* MACC-360 accumulated EPS with time. Although the green fluorescence appeared on the *C. reinhardtii* cc124 cell walls, it is likely to bind to the sugars present in cell walls. This phenomenon highlighted another difference of *C. reinhardtii* cc124 and *Chlorella* MACC-360. *Chlorella* MACC-38 did not stain with Con A, implying that it neither produced EPS nor possessed cell wall sugars with affinity for Con A (Figure 3a).

### 2.2. Effect of Microalgae Application on Plant Architecture and Canopy Cover

Pictures taken on the 45th day of growth from an aerial view allowed visualization of canopy cover in terms of area covered by green plant material, while those of 50-day-old uprooted plants showed the root structure (Figure 4). Pots in which the plant tissue appeared dense were considered to have high biomass. In contrast, pots which appeared to have sparse plant/leaf tissue were considered to have less biomass.

It seemed that the same number (20) of control plants (Figure 4a) had less canopy cover than that of algae-treated plants (Figure 4b–d). *C. reinhardtii* cc124 and *Chlorella* MACC-360-treated (Figure 4c,d) plants were profusely branched and leafier than the control (Figure 4a). *C. reinhardtii* cc124 and *Chlorella* MACC-38-treated plants had slightly more canopy cover compared to the control. The images of the uprooted plants from *C. reinhardtii* cc124 and *Chlorella* MACC-360 regimes appeared to have more leaves and axillary branches compared to the control. In addition, *C. reinhardtii* cc124 and *Chlorella* MACC-360 treatments resulted in longer roots compared to the control plants. Plants from the *C. reinhardtii* cc124 regime had the longest roots based on the images (Figure 4c); the roots from *Chlorella* MACC-360-treated samples (Figure 4d) were moderately long, while the DW/Control (Figure 4a) and *Chlorella* MACC-38-treated samples (Figure 4b) appeared to have the least root biomass.

### 2.3. Effect of Microalgae Application on Leaf Parameters

Algae application caused changes in the leaf dimensions of plants (Figure 5). The leaf dimensions measured were petiole length, leaf blade length and leaf blade width (Figure 5a–f). These were measured according to Figure 5g, following the numerical nomenclature coding of *M. truncatula* [69]. According to this nomenclature, metamers are labelled from the bottom to the top along the main axis as M1, M2, M3 and so on (Figure 5g). A metamer is the plant part including an internode, a bud and a leaf. The developmental stage of the plant parts is denoted with a decimal code from the bud stage to the fully open blue-green leaf that ranges between 0.1 to 0.9. In our study, we assessed measurements from 50-day-old plants. Consequently, all the leaves were fully matured at this time and hence the ‘0.9′ code on all measurements.

All the strains reduced the petiole length of the first true leaf. The effects on the second and third leaves were negligible. In contrast, the MA increased leaf petiole length on the older leaves (M4.9 and M5.9). *Chlorella* MACC-38 and *C. reinhardtii* cc124 effects were stronger (12–32% increase) than those of 360 (2–9% increase). Overall, none of the strains significantly affected petiole length (Figure 5a,b).

Both *Chlorella* MACC-38 and *C. reinhardtii* cc124 decreased the blade length on plants during early development. However, they increased the blade length from the third leaf onwards. In contrast, *Chlorella* MACC-360 increased leaf blade length from the first leaf onwards (Figure 5c,d). The biggest change was observed on M4.9 at 11% and the smallest on M1.9 at 2%. Strain *Chlorella* MACC-38 slightly exceeded *Chlorella* MACC-360′s effect on M4.9 and M5.9 by increasing leaf blade length by 11% and 5%, respectively. Overall, the effects of algae treatment on blade length were strong during the development period between the third and fourth leaf (Figure 5d). *Chlorella* species had stronger effects on blade length than the *Chlamydomonas* sp. Overall, *Chlorella* MACC-360 is the only strain whose effects proved to be statistically significant.

All the algae treatments had a positive impact on leaf blade width throughout the growth period except for *Chlorella* MACC-38 during late development (Figure 5e,f). This effect was more pronounced during early development where it ranged between 2% in *Chlorella* MACC-38 to 16% in *Chlorella* MACC-360-treated plants for M2.9 to M4.9. *C. reinhardtii* cc124′s effect was strongest on the fourth leaf and dropped down in subsequent leaves. Overall, *Chlorella* MACC-360 had the most pronounced effect on leaf width; it significantly increased width of the fourth, fifth and sixth leaf by 15%, 12% and 9%, respectively (Figure 5f).

Thus, the different algae strains had different effects on leaf parameters. Their effects also differed at different developmental stages. Overall, their additive effects imply that they increased the leaf size/leaf area with *Chlorella* MACC-360 showing the most striking difference, which was even visible during data collection.

### 2.4. Effect of Microalgae Application on Plant Height, Flowers, Fresh Weight, Chlorophylls and Carotenoids

In addition to leaf dimensions, we assessed more phenotypic data to capture visible differences between treatments. We measured the following physiological parameters: plant height, fresh biomass, dry biomass and flower number. We also measured chlorophyll and carotenoid levels as the representative biochemical parameters (Figure 6).

*C. reinhardtii* cc124 and *Chlorella* MACC-360 increased plant height by 2% and 11%, respectively. In contrast, *Chlorella* MACC-38 decreased plant height by 2%. Only *Chlorella* MACC-360 had a significant impact on plant height (Figure 6a).

All the algae strains increased flower number per plant by 15%, 24% and 36% in *Chlorella* MACC-38, *C. reinhardtii* cc124 and *Chlorella* MACC-360 regimes, respectively. The increase resulting from *Chlorella* MACC-360 was statistically significant (Figure 6b).

All the algae treatments increased shoot fresh weight by a range of 3% to 31%. *Chlorella* MACC-38 had the least effect with a 3% increase followed by *C. reinhardtii* cc124 with 15% and *Chlorella* MACC-360 with 31%. Only *Chlorella* MACC-360 had significantly higher shoot fresh weight compared to the control (Figure 6c).

Strains *C. reinhardtii* cc124 and *Chlorella* MACC-360 increased root fresh weight by 18% and 31%, respectively. In contrast, *Chlorella* MACC-38 decreased root fresh weight by 8% (Figure 6c).

Overall, all the algae strains increased total fresh weight, *Chlorella* MACC-38 by 4%, *C. reinhardtii* cc124 by 21% and *Chlorella* MACC-360 by 36%. Nevertheless, only *Chlorella* MACC-360’s increase was statistically significant (Figure 6c).

*Chlorella* MACC-360 treatment increased shoot dry weight (18%) and total dry weight (14%) (Figure 4d). The treatments with the other two algae strains did not significantly influence dry weight. Interestingly, all algae treatments decreased root dry weight by 9%, 14% and 18% for *Chlorella* MACC-38, *C. reinhardtii* cc124 and *Chlorella* MACC-360 treatments, respectively. Overall, none of the algae strains had a significant effect on total dry weight of plants (Figure 4d).

*C. reinhardtii* cc124 remarkably increased chlorophyll levels, showing a 32%, 35% and 32% increase in chlorophyll a, b and total chlorophyll, respectively. In contrast, *Chlorella* MACC-38 had a negligible impact on chlorophyll a (0.3% increment), a moderate effect on chlorophyll b (15% increment) and little effect on total chlorophyll (3.7% increment). In contrast, *Chlorella* MACC-360 treatment decreased both chlorophyll a and total chlorophyll by 5% and 4%, respectively, while it slightly increased chlorophyll b content by 1% (Figure 6e).

All the algae strains remarkably increased carotenoid content in plants. Interestingly, *Chlorella* MACC-38 and *Chlorella* MACC-360 extremely increased carotenoids by 31%. The effect of *C. reinhardtii* cc124 (15% increase) on carotenoid levels was half that of the chlorella strains, although it was the most significantly different from the control (Figure 6f).

Overall, *Chlorella* MACC-360 and *C. reinhardtii* cc124 strains had a more striking effect on the *M. truncatula* than *Chlorella* MACC-38 (Figure 6).

## 3. Discussion

The vast majority of studies of growth promotion in *M. truncatula* and its relatives such as *M. sativa* concentrate on the effects of growth-promoting bacteria [18,70,71,72]. These studies especially focus on the mechanisms of these microorganisms in root development and nodulation but pay little attention to the plant architecture and leaf morphology. A few studies exist on plant growth stimulating seaweed extracts applied under normal [36] and salt stressed [35] conditions. However, no study has used living green microalgae cells for growth stimulation of legumes.

Microalgae application altered the development and shoot growth of *M. truncatula* in this study compared to the control conditions. The plant height, phyllotaxy and leaf size were significantly altered in response to microalgae treatment in an algae-strain-specific manner. *Chlorella* MACC-38-treated plants had similar phyllotaxy to the control. On the contrary, bifurcation (splitting into two branches) occurred on some *C. reinhardtii* cc124-treated plants, while *Chlorella* MACC-360-treated plants had extremely enhanced axillary shoot development. *C. reinhardtii* cc124-treated plants had deformities during the vegetative phase; they lost the main shoot and developed two long branches in comparison to the control. In addition, most of the plants lacked the unifoliate leaf and flowered significantly later. These plants also had reduced axillary shoot development, a characteristic of *Headless (HDL1)* mutants. This result could be attributed to changes in the *HDL1* gene, which plays a role in the maintenance of shoot apical meristems (SAM) and leaf blade length determination. The phenotypes we observed are similar to those of *HDL 1* mutants; they have heart-shaped leaves, stems are missing (dwarf plants) and impaired flower production was observed [73]. *HDL1* was found to participate in auxin-dependent leaf morphogenesis [74] which implies that the *C. reinhardtii* cc124 treatment had an influence on auxin homeostasis in plants.

All applied microalgae species decreased leaf petiole and blade length at the early growth phase (M1.9) with the exception of *Chlorella* MACC-360, which increased blade length and width throughout the plant life. *Chlorella* MACC-38 and *C. reinhardtii* cc124 decreased blade length during juvenile stage but increased this parameter later on. Similar to *Chlorella* MACC-360, *C. reinhardtii* cc124 increased blade width throughout plant growth. In contrast, *Chlorella* MACC-38 treatment resulted in a slight decrease of blade length during the reproductive phase. Overall, all microalgae treatments increased the leaf size/area relative to control. This observation implies that the treated plants had more light-trapping capacity than the control and corresponds to the increased biomass. Another affected developmental milestone was flowering, where *Chlorella* MACC-360 induced early blooming while *C. reinhardtii* cc124 delayed flowering.

The observed results could be attributed to the possibility of differential regulation of specific plant genes involved in leaf development in *M. truncatula*. For example, *Single leaf (SGL 1)* and *fused compound leaf* (*FCL1*) genes have been found to regulate petiole development in *M. truncatula*, and mutations of both genes caused drastically reduced petioles [75]. *Phantastica (MtPhan)* has also been identified as a key regulator of petiole length. *MtPhan* suppresses *elongated petiolule* (*ELP 1)*, which is responsible for organ mortality [76]. From our phenotypic results, it seems that algae reduced expression of *MtPhan* at the very first leaf, which consequently increased that ectopic expression of *ELP1* in the rachis or petioles. Thus, the plants had reduced petiole length. It is possible that the reduced petioles could have become motor; this implies that the leaves could fold and unfold to control light intensity and reduce water loss. Thus, the young plants could be more efficient at photosynthesis than their control counterparts. The later increase in petiole length could be a strategy to reach out into open space to avoid shade as the plants become bushy. Overall, it is likely that algae treatment interfered with the expression of genes involved in petiole development especially the *MtPhan*. Also notable is the fact that development of the first unifoliate leaf was completely aborted, and some leaflets were occasionally mismatched in most of the algae-treated plants irrespective of strain. This could be linked to the possible interference with the expression of *MtPIN 10*, an auxin efflux transporter that plays a critical role in dissected leaf and flower development [76].

*Stenofolia (STF)* is another gene involved in leaf and floral lateral development [77]. Because the leaves from algae-treated plants had altered leaf dimensions in comparison to the control, it is possible that they had a similar effect on this gene. Just like *HDL1*, *STF* regulates leaf growth by controlling auxin levels [78,79,80]. Current literature suggests that *STF* modulates auxin and cytokinin homeostasis as well the hormonal crosstalk that coordinates developmental signals at the adaxial–abaxial interface of leaf primordial [81].

Microalgae application increased plant height in all cases, although only the increase resulting from *Chlorella* MACC-360 was significant. The plant biomass and flower number also showed a clear increase. Another significant observation was the remarkable increase in pigments (chlorophylls and carotenoids). These results are consistent with previous studies on the effect of biostimulants on both monocot and dicot plants [5,41,47,51,82,83,84].

The enlarged leaves in algae-treated plants could be due to the enhanced cell division and cell elongation by the phytohormones in the MA treatments. Microalgae application could be expected to directly impact shoot and root elongation since eukaryotic green microalgae produce auxins and cytokinins. Our results are consistent with what has been reported on auxin producing microorganisms such as bacteria and endophytic fungi, which were found to promote plant growth [18,70,71,72,85,86,87,88]. The difference observed in the different algae treatments could be attributed to the ratios of auxins/cytokinins or different concentrations of specific phytohormones. One particular study revealed that even strains from the same genus could have huge differences in cytokinin production [89]. Moreover, various microalgae might produce different types of auxins and cytokinins as well as other hormones, which were not quantified in this study. Stirk [66] showed that most algae strains produce the indole-3-acetic acid (IAA) form of auxin in higher proportions than the indole-3-acetamide (IAM). The same author goes on to reveal that three different forms of cytokinins are prevalent in microalgae implying differential hormone producing capacities among strains. All the same, *Chlorella* MACC-360 was found to endogenously produce a plethora of plant-growth-promoting phytohormones [90].

The cell size of the microalgae might also play a significant role in the interaction with other microbes and plant surface at the root interphase. Small size could mean that the algal cells fit in a smaller space and interact with more microbes and larger plant surface. Thus, based on cell size, it is possible to hypothesize that *Chlorella* MACC-360 has more interactions/contacts with other microbes and with the plants in the soil than *Chlorella* MACC-38 and *C. reinhardtii* cc124.

Exopolysaccharides (EPS) released by one of the applied microalgae strains could explain the pronounced effect of *Chlorella* MACC-360 on plants in comparison to the other two algae. The presence of a significant number of extracellular polysaccharides in the vicinity of *Chlorella* MACC-360 implies that this strain has the capacity to alter its immediate environment. The secreted metabolites might either attract or repel microorganisms and trigger biological responses from organisms including the plants. The material could also aid in water and air circulation in the soil. Consequently, MA influence the physical, chemical and the biological properties of the rhizosphere.

EPS have been found to be indispensable in rhizobium–legume symbiosis and hence in nitrogen fixation. The presence of algal EPS could help in recruitment of beneficial bacteria and fungi to the plant rhizosphere. Furthermore, the presence of EPS in soil has also been found to improve soil drainage and nutrient availability by increasing the content of ions in soil [91]. EPS isolated from a *Chlorella* species were found to possess immunomodulatory and antioxidant capacity too; the former refers to modulating biological response, while the latter implies a role in response to reactive oxygen species (ROS) [92]. These properties imply that EPS participate in designing microbial interactions and response to stress. Furthermore, EPS contain sugars, which the plants may directly absorb and use for growth. Studies to evaluate the biostimulatory effect of *Chlorella* derived polysaccharides showed that they improve plant growth, pigment content and fresh biomass [93].

EPS also strongly facilitate biofilm development. Biofilms have been identified to contribute towards growth promotion in plants. Biofilms improve soil characteristics by absorbing atmospheric moisture and trapping water in the topsoil layers thus making it more available to plants especially in sandy soils. They also reduce water infiltration and hence prevent soil erosion [94]. Biofilmed biofertilizers (BFBFs) prepared with microbial consortia have proved to be a sustainable means of increasing crop yield [95]. Addition of cyanobacteria into areas undergoing desertification was found to stimulate biocrust formation, which improved soil properties and triggered plant succession [96]. In an independent study, biocrusts were found to reduce loss of soil organic carbon content via soil erosion [97]. These studies revealed the role of biofilm forming microorganisms in maintenance of soil fertility and their potential as tools for soil resources conservation and restoration of fertility to dry land. *Chlorella MACC*-360 is a strong EPS producer and can form biofilms as evidenced by the cell aggregations shown in the microscopy pictures. This phenomenon could explain the observed strong growth promotion effects it exerted on *M. truncatula*.

## 4. Materials and Methods

### 4.1. Algae Strains

Two microalgae species belonging to the *Chlorella* genus (*Chlorella* MACC-360 and *Chlorella* MACC-38) taken from the Mosonmagyaróvár Algal Culture Collection (MACC) and the *Chlamydomonas reinhardtii* cc124 strain were selected for plant biostimulant studies based on their rapid biomass accumulation.

#### 4.1.1. Determination of Algal Growth

Under aseptic conditions, the surface of a 7-day lawn algae culture in a tris-acetate-phosphate (TAP)-agar plate was scrubbed with a sterile rod and dipped into a 10 mL falcon tube containing 5mL of TAP media. The mouth of the falcon tube was flamed before capping. The tubes were placed in an incubator with the following conditions; 25 °C, 16/8 h light/dark regime, white light and shaker set at 180 rpm. After 3 days, the cultures’ optical density was determined by measuring absorbance at 750 nm with a spectrophotometer. The cultures were then used for inoculation into 1500 μL TAP media in 24-well plates to make cultures with a final optical density (O.D) of 0.2 at 750 nm. Each strain was replicated 6 times. A blank was also maintained and replicated 6 times as well. The plate was placed in the incubator, and optical density was determined by a Hidex plate reader once per day.

#### 4.1.2. Determination of Cell Numbers

Three-day-old starter cultures were inoculated into 25 mL of TAP media to an initial O.D of 0.2. Two flasks per strain were prepared. The flasks were incubated in the same incubator described above. To determine the cell numbers, 100 μL was drawn from the flasks and made up to a volume of 1 mL using water. Then, 10 μL of the diluted culture was placed in the Luna slides, and cell numbers were determined with the fluorescent algae protocol in the Luna Automated cell counter (Luna FL-Logos Biosystems). Cell counts were determined once at the same time every day for 5 days. Cell numbers per day from each flask were individually plotted using GraphPad Prism.

### 4.2. Microscopy

#### 4.2.1. SEM—Scanning Electron Microscopy

Eight μL of algae samples was spotted onto a silicon disc coated with 0.01% (*w*/*v*) poly-l-lysine (Merck Millipore, Billerica, MA, USA). Cells were fixed with 2.5% (*v*/*v*) glutaraldehyde and 0.05 M cacodylate buffer (pH 7.2) in PBS overnight at 4 °C. The discs were washed twice with potassium buffered saline (PBS) and dehydrated with a graded ethanol series (30%, 50%, 70%, 80%, 100% ethanol (*v*/*v*), for 1.5 h each at 4 °C). The samples were dried with a critical point dryer, followed by 12 nm gold coating (Quorum Technologies, Laughton, East Sussex, UK) and observed under a JEOL JSM-7100F/LV scanning electron microscope (JEOL Ltd., Tokyo, Japan).

#### 4.2.2. CLSM—Confocal Laser Scanning Microscopy

From the flasks with cultures, 50 μL was drawn out into an Eppendorf tube and stained with CFW and Con A both at a concentration of 10 µg/μL. After 30 min incubation in dark, the cells (8 μL) were spotted on microscope slides and covered with 2% (*w*/*v*) agar slices and observed with an Olympus Fluoview FV 1000 confocal laser microscope with 60× magnification objective. Sequential scanning was used to avoid crosstalk of the fluorescent dyes and chlorophyll autofluorescence.

### 4.3. Preparation of the Algae for Plant Treatment

Broth cultures of the algae strains in TAP media, pH 7, were cultivated for preparation of plant treatment. Under aseptic conditions, the surface of a fully grown lawn algae culture from a TAP-agar plate was scrubbed with a sterile rod. The rod was dipped in a 50 mL Erlenmeyer flask containing 15 mL of TAP media. The mouth of the flask was flamed before capping. The flasks were then placed in an algae growth chamber with the following conditions: 25 °C, 16/8 h light/dark regime, white light and shaker set at 180 rpm. After 5 days, 5 mL of the culture was transferred into a 100 mL Erlenmeyer flask containing 50 mL TAP media and placed in the aforementioned growth conditions. The cultures were left to grow for 7 days. On the 7th day, 5 mL of the culture was transferred into a new conical flask containing 50 mL of TAP media to start culture for the next application. The remaining 50 mL was used to prepare the algae treatment.

The 7-day-old cultures were transferred into 50 mL falcon tubes and centrifuged at 4600 rpm for 15 min. The supernatant was discarded, and the cells were resuspended in 50 mL of sterile distilled water. The suspension was centrifuged again at 4600 rpm for 15 min, and the supernatant was discarded. The pellet was then resuspended in sterile distilled water at a concentration of 1 g/L. Total carbon and nitrogen content of the 1 g algae pellet was determined using an elemental analyzer. Control solution (referred to control/distilled water throughout the manuscript) contained respective amounts of sodium acetate and ammonium-chloride dissolved in distilled water.

### 4.4. Greenhouse-Based Bioassays

*Medicago truncatula* seeds of line A17 Jemalong, were used for the studies. Planting and phenotyping were done according to Bucciarelli [69]. Plants were surface scarified with concentrated sulfuric acid for 5 min and thoroughly washed with sterile ice-cold water. Seeds were consequently surface sterilized for 3 min with 0.01% HgCl_2_ and then washed 5 times with sterile distilled water. The seeds were allowed to germinate for 2 days at 4 °C then transferred to Petri dishes with moistened filter paper and given a 21-day vernalization period at 4 °C. The plates were then transferred to a growth chamber for 2–3 days.

Vernalized seeds with a radical length of 1 to 1.5 cm were treated with each of the treatments (distilled water/DW, *Chlorella* MACC-38, *C. reinhardtii* cc124 and *Chlorella* MACC-360 algae suspensions) for 20 min. The seedlings were then rinsed with water and planted in pots containing soil mixed with vermiculate in the ratio of 3:1. Pot size was 10 × 10 × 35 cm^3^. Each pot contained 4 plants, and each treatment had 5 pots placed in one box. Plants were fertilized during transplantation with 100mL of Solution I (Sol I) diluted 40× from the stock solution prepared as follows: First, the following macronutrient stock solutions were prepared separately: 20.2 g/L KNO_3_, 73 g/L CaCl_2 ×_ 2H_2_O, 24.6 g/L MgSO_4_, 43.5 g/L K_2_SO_4_, 8.2 g/L Fe-Na-EDTA and 27.2 g/L KH_2_PO_4_ and 0.05 M H_3_BO_3_. Secondly, a microelement stock solution was prepared by adding 6.2g MnSO_4_, 10 g KCl, 1 g ZnSO_4 ×_ 7H_2_O, 1g (NH_4_) Mo_7_O_2 ×_ 4H_2_O, 0.5 g CuSO_4_ and 0.5 mL H_2_SO_4_ into water and topping it up to 1 L. The stock solutions and 800 mL distilled water were autoclaved separately. Finally, 25 mL of each of the macronutrient solutions and 1.35 mL micronutrient stock solution were added into 800 mL of sterile water to make Sol 1 stock solution. Plants were grown in the greenhouse at 24 to 26 °C and with a 16 h photoperiod. Plants were watered weekly with the water-based algae suspensions (0.05 g/L) for the algae regimes and control solution for the control/distilled water regime, with the last treatment being on the 35th day of growth.

After 45 days, aerial pictures of the plants were taken with a camera to capture the plant cover. After 50 days, the growth experiments were terminated. Ten plants from each treatment were used for phenotyping. Leaf parameters were measured for the leaves associated with the first to the fifth metamer (M1–M5) for leaf petiole and leaf blade length. Leaf blade width was measured up to the sixth metamer (Figure 5g). Leaf blade length was measured along the midrib, from the tip of the middle leaflet to the end of leaf petiole. Leaf blade width was determined as the distance between the two opposite leaflets of a trifoliate leaf. Plant height (height of main axis or one of the axis in bifurcated plants) and flower number were also recorded. All measurements were taken with a flexible handheld ruler.

Plants were gently uprooted and the roots thoroughly washed with distilled water to remove all the soil debris. Five plants were laid out on a black background and pictures taken with a camera. Shoots and roots were separated and weighed separately to record fresh biomass. The plants were then dried in a dry air oven for 48 h at 70 °C, and the dry weight was recorded. Dry weight was recorded as an average of the pooled sample for each treatment regime.

Another set of 10 plants per treatment was collected and processed for determination of plant pigment content determination. For each treatment, 2 pooled samples from 5 plants were put into separate tubes. About 0.1 g of this fresh leaf material was placed in a test tube, and 10 mL of 80% acetone was added. The tubes were placed in a water bath set at 60 °C for 30 min and cooled in ice. Then, 200 μL of the extract was transferred into two wells in a 96-well plate, and absorbance values were measured with a HIDEX plater reader. The content of chlorophylls was calculated according to Arnon equations, and the formula for carotenoids was adopted from Lichtenthaler et al. equation specific for acetone extracts [98,99].

### 4.5. Statistical Analysis

Data from 3 independent experiments were used for statistical analysis; the data represented parameters measured from 30 plants from each treatment and in total 120 plants. The collected data were tested for normality and homoscedasticity. Multiple comparisons of the groups or treatments were performed with analysis of variance (one-way ANOVA) for plant height, flower number, biomass and pigment parameters. Two-way ANOVA was applied for the comparison of leaf parameters data, which were in the format of grouped data. Tukey’s multiple comparison test with the alpha 0.05 was used to analyze the significance of differences. All statistical analyses were executed using GraphPad Prism 8.

## 5. Conclusions

The tested eukaryotic green microalgae exerted growth stimulating effects on *Medicago truncatula*, a phenomenon attributable to phytohormones and algal EPS production. The algae application on plants influenced leaf size, biomass accumulation, pigment content and pod/flower production. *Chlorella* MACC-360 had the most significant impact on *Medicago* plants. However, the treatment with *C. reinhardtii* cc124 persistently increased both chlorophyll and carotenoid contents of the plant contrary to the applied *Chlorella* species. These results inspire insightful studies to elucidate the mechanism of the different microalgae on plants at the molecular level.

Future studies will employ microscopy techniques to elucidate the status of the interaction between the microalgae and plant roots. The differential expression of the described genes will also be studied via targeted molecular techniques such as quantitative polymerase chain reaction (qPCR) to obtain better insight into the effects of microalgae treatments on plants at the molecular level.

## Figures and Tables

**Figure 1 plants-10-01060-f001:**
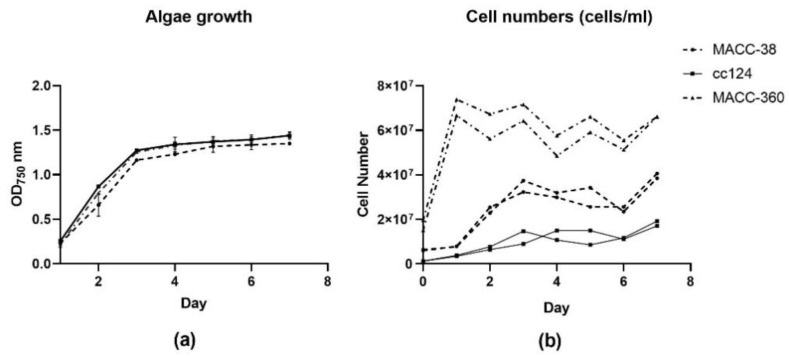
Growth parameters showing optical density and cell numbers of the algae strains grown under light/dark conditions over 7 days: (**a**) growth curve; (**b**) cell numbers. MACC-38 = *Chlorella* MACC-38; cc124 = *C. reinhardtii* cc124 and MACC-360 = *Chlorella* MACC-360 on both graphs.

**Figure 2 plants-10-01060-f002:**
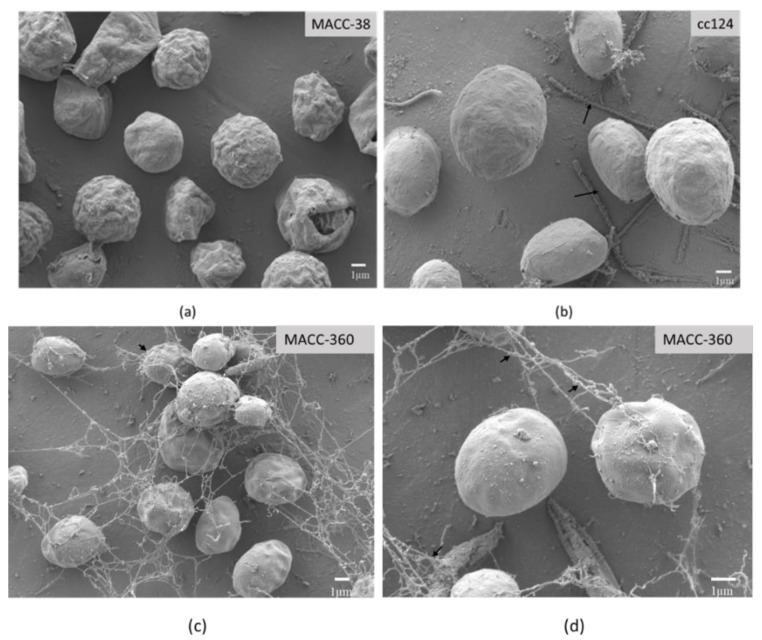
Scanning electron microscopy pictures of the three strains: (**a**) *Chlorella* MACC-38; (**b**) *C. reinhardtii* cc124; (**c**) *Chlorella* MACC-360 at 5000× magnification; (**d**) *Chlorella* MACC-360 at 10,000× magnification. Black arrows on (**b**) show flagella; black arrows on (**c**) show aggregation or clustering of cells while black arrows on (**d**) show the extracellular material connecting one cell to another in the cell aggregations/matrix.

**Figure 3 plants-10-01060-f003:**
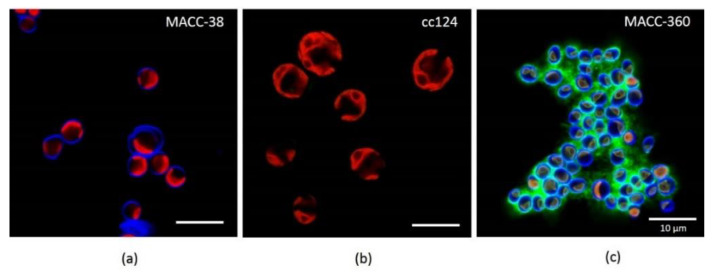
CLSM pictures of live microalgae cells on the 7th day after inoculation; (**a**) *Chlorella* MACC-38; (**b**) *C. reinhardtii* cc124; (**c**) *Chlorella* MACC-360 stained with calcofluor white (CFW) and concanavalin A (Con A). The blue fluorescence is CFW dye, which stains the cell walls, red is the chloroplast autofluorescence of live cells and green fluorescence is Con A dye, which binds to extracellular polysaccharides (EPS).

**Figure 4 plants-10-01060-f004:**
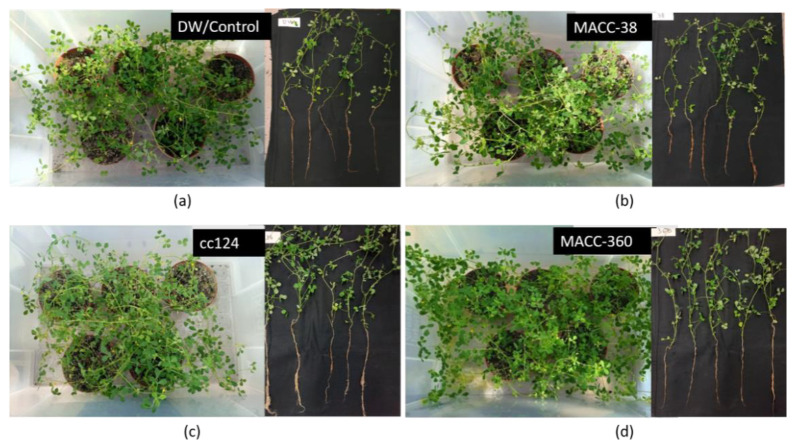
Aerial pictures of the plants (five pots per treatment placed in a box) at 45 days after planting: (**a**) DW/Control plants; (**b**) *Chlorella* MACC-38-treated plants; (**c**) *C. reinhardtii* cc124-treated plants and (**d**) *Chlorella* MACC-360-treated plants. For all figure panels, the left image shows the aerial view, while the right one shows the front view of uprooted plants.

**Figure 5 plants-10-01060-f005:**
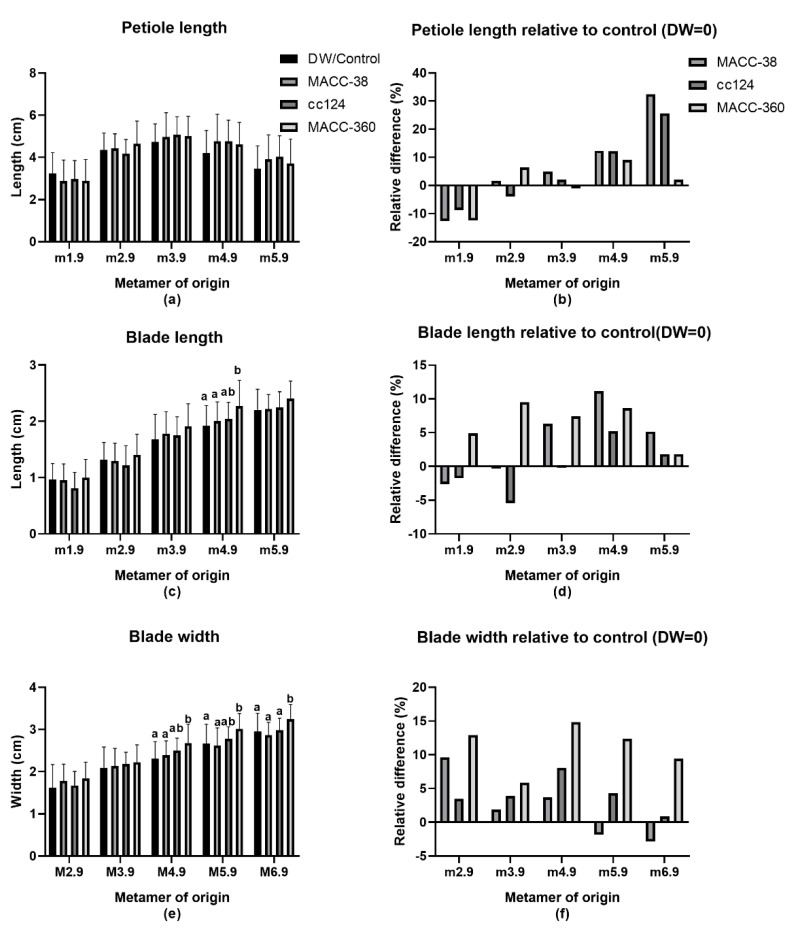
Effect of algae application on the leaf size of 50-day-old *Medicago truncatula* leaves; 10 replicates were measured per experiment. (**a**) Leaf petiole; (**b**) petiole length relative to control; (**c**) leaf blade length; (**d**) leaf blade length relative to control; (**e**) leaf blade width; (**f**) leaf blade width relative to control; (**g**) Illustrative diagram of *M. truncatula,* created with BioRender.com, showing the different phenotypic parameters that were measured. Metamers (internode, leaf and bud) and their associated leaves are labelled from the bottom to the top along the main axis in ascending order. The red arrow on the first leaf depicts blade width, the dark blue arrow depicts blade length and the brackets show the petiole length. Different letters on the bars indicate significant differences between groups (*p* < 0.05) according to Tukey’s multiple comparison test. Two-way ANOVA was used for all parameters.

**Figure 6 plants-10-01060-f006:**
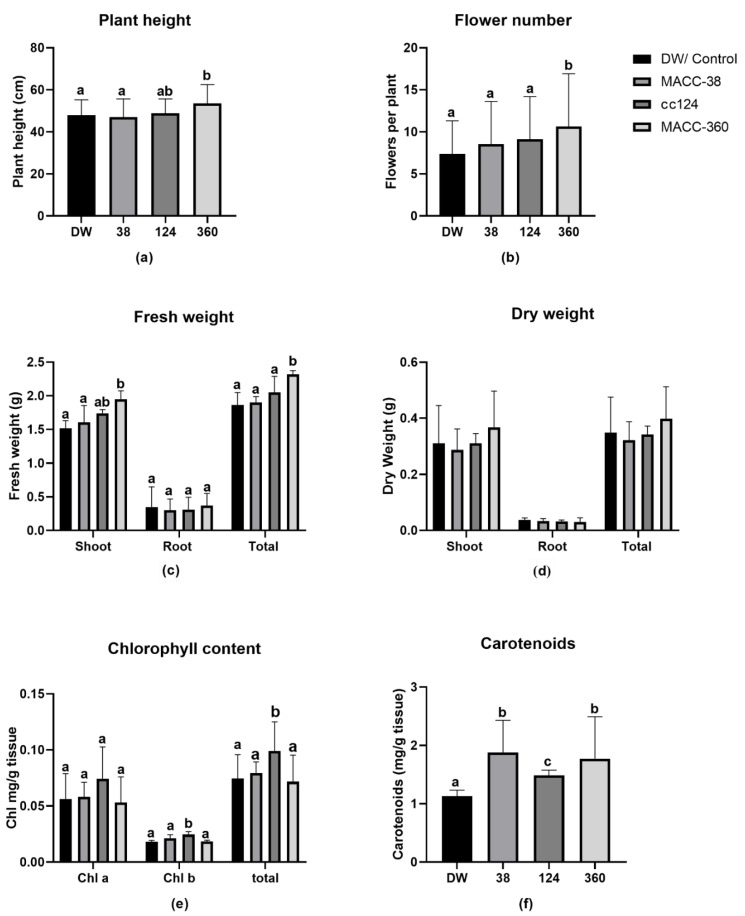
Effects of microalgae applications on plants (50-day-old plants); (**a**) plant height; (**b**) flower number; (**c**) fresh weight; (**d**) dry weight; (**e**) chlorophyll; (**f**) carotenoids. Data represent means and standard errors (error bars) of 10 biological replicates per experiment. Different letters on bars indicate significant differences between groups (*p* < 0.05), according to Tukey’s test. One-way ANOVA was used for all parameters.

## Data Availability

The data presented in this study are available in the main text and in Appendix A.

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
