# Peer review of "Strain-Specific Biostimulant Effects of Chlorella and Chlamydomonas Green Microalgae on Medicago truncatula"

_plants, 2021, doi:10.3390/plants10061060_

Round 1

Reviewer 1 Report

The paper focus on the use of living microalgal cells as biostimulant drenching application to Medicago to enhance their growing performances.

The study is carried out carefully in relation to the methods used. It may be improved by applying microscopy to reveal the intimate interaction between microalgae and plant roots. Moreover, molecular investigations may help clarify the hypothetical mechanisms that are cited in the discussion. Please, report these considerations as future perspectives.

Moreover, can you include photographs of root and shoot architecture (front view) for each treatment? Please include dry weigth graph, also.

Author Response

Q1. The study is carried out carefully in relation to the methods used. It may be improved by applying microscopy to reveal the intimate interaction between microalgae and plant roots. Moreover, molecular investigations may help clarify the hypothetical mechanisms that are cited in the discussion. Please, report these considerations as future perspectives.

Thank you for these valuable suggestions. We fully agree that is highly important to get deeper insight into the interaction between plant roots and microalgae. We intend to apply both advanced microscopy and molecular approaches (whole community transcriptome analysis followed by targeted qPCR validation). We have included these in the conclusion section of the revised manuscript as our future perspectives.

Q2. Moreover, can you include photographs of root and shoot architecture (front view) for each treatment? Please include dry weigth graph, also.

Thank you for making these suggestions. We have provided the pictures and dry weight data as suggested.

Reviewer 2 Report

The manuscript entitled “Strain-specific biostimulant effects of Chlorella and Chlamydomonas green microalgae on Medicago truncatula” by Gitau et al. includes most of the recent research interest carried out in the field of biofertilizers which is appreciable. Answered most of the questions on algal biofertilizers. This manuscript can be accepted with minor revision.

Minor suggestions:

  • The introduction seems to be lengthy, needs to be more crisp while reading for a viewer, authors may need to keep more focus on the hypothesis-related introduction.
  • Line 35 and line 53: Why only phosphorus and nitrogen has been discussed, whether it had any relation to materials and method section or results in the manuscript (mean to say any experimental findings on this specific trait in this manuscript?). Please explain.
  • Lines 136-139: What might be the reason for the difference in cell number between both samples. Please describe your observations.
  • Line 187: Please write this in the footnotes of Figure 4.
  • Lines 19-193 describing the methods. Please focus more on your findings instead of repeating the methods in the results section.
  • Line 199: "the same number of control plants" - give an accurate number here.
  • Line 369 to 374: While citing the examples of works for correlation or consistency with specific work, authors need to correlate the result of this work with the other's research. Instead of a hypothetical view, an example is, auxin production of algae, please cite an example where the same species produced Auxin to make it more authentic instead of saying most algae produce auxins (line 373).
  • Also, I would suggest thorough proofreading of the manuscript for rectifying grammatical and usage errors if any. A lot of type errors found throughout the manuscript. Space is missing between words, digits, units, etc. Please rectify.

This manuscript has included most of the relevant literature available but there a few minor suggestions to be made. I would recommend the publication of this manuscript after addressing minor changes.

Author Response

The manuscript entitled “Strain-specific biostimulant effects of Chlorella and Chlamydomonas green microalgae on Medicago truncatula” by Gitau et al. includes most of the recent research interest carried out in the field of biofertilizers which is appreciable. Answered most of the questions on algal biofertilizers. This manuscript can be accepted with minor revision.

Minor suggestions

Q1. The introduction seems to be lengthy, needs to be more crisp while reading for a viewer, authors may need to keep more focus on the hypothesis-related introduction.

Thank you for your suggestion. We have significantly shortened and focused the introduction. We have deleted the general descriptions and comparison of biofertilizers and biostimulants.

Q2. Line 35 and line 53: Why only phosphorus and nitrogen has been discussed, whether it had any relation to materials and method section or results in the manuscript (mean to say any experimental findings on this specific trait in this manuscript?). Please explain.

Phosphorous and nitrogen are the most common limiting factors to plant growth as well as the major constituents of most chemical fertilizers. The excessive use of these fertilizers is detrimental to the whole ecosystem since a large portion ends up in the water bodies causing serious eutrophication. We just wanted to call attention for this environemnatl issue in the introduction. Here we did not conduct experiments to test the specific supply effects of MA concerning these two nutrients in the soil, it was not among our specific goals.

Q3. Lines 136-139: What might be the reason for the difference in cell number between both samples. Please describe your observations.

The main reason is the difference in cell size among the three strains.

We used optical density (OD) determination for standardizing the experimental setup. Due to the large difference in cell size, the strains with bigger cell size had less cells at the same OD compared to the strains with smaller cell size. This is clear with the initial cell numbers where Chlorella MACC-360 starts off with a higher cell number than Chlorella MACC-38 at the same OD. The same applies to Chlamydomonas reinhardtii cc124.

Besides cell size, the differences can be attributed to strain-specific cell division cycle rate (growth rate). Chorella MACC-360 has a shorter cell cycle (higher growth rate) than Chlorella MACC-38.

We have added this information in the revised manuscript.

Q4. Line 187: Please write this in the footnotes of Figure 4.

Thank you for your suggestion. We have made the change accordingly.

Q5. Lines 19-193 describing the methods. Please focus more on your findings instead of repeating the methods in the results section.

Thank your for your comment. We have removed all information that describes the methodology from this section and only focused on the results.

Q6. Line 199: "the same number of control plants" - give an accurate number here.

Thank you. We have specified it as 20 plants per treatment.

Q7. Line 369 to 374: While citing the examples of works for correlation or consistency with specific work, authors need to correlate the result of this work with the other's research. Instead of a hypothetical view, an example is, auxin production of algae, please cite an example where the same species produced Auxin to make it more authentic instead of saying most algae produce auxins (line 373).

Thank you for your valuable suggestion. We have added a reference in which Chlorella MACC-360 was studied and found to produce a plethora of phytohormones. Stirk WA, Bálint P, Tarkowská D, Novák O, Maróti G, Ljung K, Turečková V, Strnad M, Ördög V, Van Staden J. Effect of light on growth and endogenous hormones in Chlorella minutissima (Trebouxiophyceae). Plant Physiology and Biochemistry. 2014 Jun 1;79:66-76.

Q8. Also, I would suggest thorough proofreading of the manuscript for rectifying grammatical and usage errors if any. A lot of type errors found throughout the manuscript. Space is missing between words, digits, units, etc. Please rectify.

Thank you for pointing out this. We have thoroughly proofread the manuscript and made the necessary corrections where necessary.

This manuscript has included most of the relevant literature available but there a few minor suggestions to be made. I would recommend the publication of this manuscript after addressing minor changes.

Reviewer 3 Report

The manuscript entitled" Strain-specific biostimulant effects of Chlorella and Chlamydomonas green microalgae on Medicago truncatula is an interesting study conducted by the authors.

The growth parameters were assessed on application of biostimulant, which is need of the hour, however I have one major concern regarding the growth parameter, that the dry biomass measurement was not provided for the experiment which is highly recommended for growth yield experiments as fresh biomass has huge variability due to higher water content–

  • Abstract- Authors claim biostimulant have positive impact on studied plant but only mentioned auxiliary parameters, didn't mention about increment in biomass yield which is the main concept behind the study—please correct and incorporate few sentences for biomass yield ----
  • Authors have not provided the dry herb yield after the termination off the experiment, which is highly desired for growth yield experiments – please provide the same if possible, or give reasons for not including the data.
  • Fig 6c, authors represented data for shoot weight followed by root weight and then total, why total was provided, its not clear, secondly what is compared to what is not clear with the figures, its better if Duncan letters van be used instead of stars so that readers can find out what is compared, and how they are significantly different.
  • Similarly with other figures too, wherever stars are provided it is advisable to please use stastical letters to present the significant differences as it is hard to understand form the present format---
  • In some figures authors have used indications as MACC-360, MACC-38, while somewhere it is 38, 360, please follow at above pattern as the number itself is confusing and authors also must follow a single pattern of writing---
  • Research Gap is not well presented, if possible please refine the sentences—
  • Material and methods – the research design needs to be presented more clearly which is lacking in the present manuscript—
  • Section 4.1.1 and 4.3 seems same process is described again and again, if I have understood correctly. Please check and if needed reframe---
  • There is no mention of how and when the experiment was terminated and what were studied parameters—

Author Response

We would like to thank the Reviewer for the time spent on our manuscript. We have made the revisions following the suggestions and trust that the changes are acceptable.

Q1. The growth parameters were assessed on application of biostimulant, which is need of the hour, however I have one major concern regarding the growth parameter, that the dry biomass measurement was not provided for the experiment which is highly recommended for growth yield experiments as fresh biomass has huge variability due to higher water content.

Thank you for your comment. You are absolutely right about the biomass parameter. We have provided dry weight data in the revised manuscript.

Actually we intentionally omitted these data from the original manuscript version since we recorded the dry weight from a pooled sample and calculated the average values. Thus, we only had 3 data points per treatment which means that we were not able to statistically determine the differences (we lost the variation in groups by using pooled samples of 10 plants per treatment). We did so because plant parts from individual plants mixed up during the drying process or parts fell out from individual plants and there was no way to tell which part fell from which individual plant. Also, these plants were fully grown (50 days) and very large, hence difficult to handle individually when dried.

Q2. Abstract- Authors claim biostimulant have positive impact on studied plant but only mentioned auxiliary parameters, didn't mention about increment in biomass yield which is the main concept behind the study—please correct and incorporate few sentences for biomass yield.

Thank you once again for focusing on this important parameter. We have added the information in our abstract.

Q.3 Authors have not provided the dry herb yield after the termination off the experiment, which is highly desired for growth yield experiments – please provide the same if possible, or give reasons for not including the data.

We have provided this data. Please see answer to Q.1 above.

Q4. Fig 6c, authors represented data for shoot weight followed by root weight and then total, why total was provided, its not clear, secondly what is compared to what is not clear with the figures, its better if Duncan letters van be used instead of stars so that readers can find out what is compared, and how they are significantly different.

Thank you for your detailed review of the Figures and the statistical analyses. We have repeated the statistical analyses and used Two-way ANOVA for the leaf parameters (grouped data) and One-way ANOVA for all other parameters, as well as Tukey’s multiple comparison tests in all cases. We have also replaced asterisks with letters as per your suggestions.

We showed both individual (shoot and root) weight parameters and the total weight since we collected these data separately and the separate data were informative. Separation of the two plant parts enabled recognition of the specific actions of MA on plants, significant biostimulation was observed only on the shoot part in our experiments. This information might be valuable for application in agriculture for different plants depending on the part which is of economic value. From this same perspective, fresh weight is also an important parameter as much as dry weight is.

Q5. Similarly with other figures too, wherever stars are provided it is advisable to please use stastical letters to present the significant differences as it is hard to understand form the present format

Thank you for helping us improve our results representation. We have made the adjustments accordingly.

Q.6 In some figures authors have used indications as MACC-360, MACC-38, while somewhere it is 38, 360, please follow at above pattern as the number itself is confusing and authors also must follow a single pattern of writing

Thank you for your suggestions, we have made corrections where necessary. We have even changed the figure legends in all figures except the supplementary figure in which we have provided a detailed annotation.

Q.7 Research Gap is not well presented, if possible please refine the sentences

This statement defines the research gapHowever, to the best of our knowledge, no study sought to elucidate the effects of axenic monocultures of Chlorophyta microalgae without any other accompanying microbes on M. truncatula plants’.

Firstly, no such study on M. truncatula has been done. Secondly, biostimulant studies mostly focus on Chlorella species while our study also incorporated a Chlamydomonas species. Thirdly, most studies use extracts or combined mixed cultures of the algae and partner bacteria. Our study used whole algae cells of axenic cultures. All these details are outlined in our manuscript.

Q.8 Material and methods – the research design needs to be presented more clearly which is lacking in the present manuscript

Thank you for your valuable comment. We have added more information in the methodology section to make it more clear. We have also added information on how phenotyping was done and consequently added results for biomass as well as plant pictures. We have also clearly explained how we did statistical analysis. We trust that it is now acceptable.

Q.9 Section 4.1.1 and 4.3 seems same process is described again and again, if I have understood correctly. Please check and if needed reframe

Thank you for your comment. We have clarified this by providing clear subtitles. Section 4.1.1 was changed for „Determination of algal growth”. These assays were done on 24-well plates. The only similarities with section 4.3 are the growth media and the applied growth incubator.

Section 4.3 provides a detailed methodology of how MA cultures were prepared for use as biostimulants. The cultures were maintained in flasks and were only removed on day 7 for centrifugation and resuspension before being applied on plants.

Q.10 There is no mention of how and when the experiment was terminated and what were studied parameters

Thank you pointing out that this was not clear. We maintained the plants in the greenhouse for 50 days and applied treatment on weekly basis. This means we provided 6 treatments. Thus, we did not apply algae treatment on last week of plant growth (day 49) since we terminated after 50 days of growth. We have clarified this in the manuscript.

Round 2

Reviewer 3 Report

The comments and suggestions on the manuscript entitled" Strain-specific biostimulant effects of Chlorella and Chlamydomonas green microalgae on Medicago truncatula were satisfactorily and well addressed by the authors. It's nice to see that Dry weight has been added to the MS.

Thanks for the revision and for addressing the comments for the overall improvement of the manuscript.